# Exercise Training and Vitamin C Supplementation Affects Ferritin mRNA in Leukocytes without Affecting Prooxidative/Antioxidative Balance in Elderly Women

**DOI:** 10.3390/ijms21186469

**Published:** 2020-09-05

**Authors:** Małgorzata Żychowska, Agata Grzybkowska, Monika Wiech, Robert Urbański, Wanda Pilch, Anna Piotrowska, Olga Czerwińska-Ledwig, Jędrzej Antosiewicz

**Affiliations:** 1Department of Sport, Faculty of Physical Education, Kazimierz Wielki University in Bydgoszcz, 85-064 Bydgoszcz, Poland; 2Department of Biochemistry, Faculty of Physical Education, Gdansk University of Physical Education and Sport, 80-336 Gdansk, Poland; agata.p3@gmail.com; 3Department of Health Promotion, Faculty of Tourism and Recreation, Gdansk University of Physical Education and Sport, 80-336 Gdansk, Poland; wiechmonika3@gmail.com; 4Department of Biomechanics and Sports Engineering, Faculty of Physical Education, Gdansk University of Physical Education and Sport, 80-336 Gdansk, Poland; robert.urbanski@awf.gda.pl; 5Department of Cosmetology, Faculty of Physiotherapy, University of Physical Education in Krakow, 31-571 Krakow, Poland; wfpilch@poczta.onet.pl (W.P.); anna.piotrowska@awf.krakow.pl (A.P.); olga.malgorzata.czerwinska@gmail.com (O.C.-L.); 6Department of Bioenergetics and Exercise Physiology, Faculty of Health, Medical University of Gdansk, 80-210 Gdansk, Poland

**Keywords:** supplementation, ageing, ferritin gene expression, FOXO3a, catalase, oxidative stress

## Abstract

Physical training and antioxidant supplementation may influence iron metabolism through reduced oxidative stress and subsequent lowering of mRNA levels of genes that are easily induced by this stress, including those responsible for iron homeostasis. Fifteen elderly women participated in our 12-week experiment, involving six weeks of training without supplementation and six weeks of training supported by oral supplementation of 1000 mg of vitamin C daily. The participants were divided into two groups (*n* = 7 in group 1 and *n* = 8 in group 2). In group 1, we applied vitamin C supplementation in the first six weeks of training, while in group 2 during the remaining six weeks of training. In both phases, the health-related training occurred three times per week. Training accompanied by vitamin C supplementation did not affect prooxidative/antioxidative balance but significantly decreased ferritin heavy chain *(FTH*) and ferritin light chain (*FTL*) mRNA in leukocytes (for *FTH* mRNA from 2^64.24 to 2^11.06, *p* = 0.03 in group 1 and from 2^60.54 to 2^16.03, *p* = 0.01 in group 2, for *FTL* mRNA from 2^20.22 to 2^4.53, *p* = 0.01 in group 2). We concluded that vitamin C supplementation might have caused a decrease in gene expression of two important antioxidative genes (*FTH, FTL)* and had no effect on plasma prooxidative/antioxidative balance.

## 1. Introduction

Oxidative stress can influence iron metabolism and iron-regulatory protein status [1]. Genes associated with cellular stress response and iron metabolism, such as *FTH1* (ferritin heavy chain 1)*, FTL* (ferritin light chain), *PCBP1* (poly(rC)-binding protein 1), *PCBP2* (poly(rC)-binding protein 2), *FOXO3a* (forkhead box O3A), or *CAT* (catalase) are easily induced by oxidative stress. Oxidants can directly induce ferritin gene expression by targeting conserved regions of these genes [2] as well as by regulating the activity of mRNA at transcriptional levels, and proteins at post-transcriptional levels [3]. For example, inactivation of iron-regulating protein 1 (IPR1) by oxidative stress can be associated with not blocking ferritin mRNA [4]. Moreover, expression of these genes is directly regulated by cellular iron status.

All genes investigated in this study are involved in iron homeostasis and are easily induced by stressors. *FTH* and *FTL* encode “acute phase” proteins. It is well known that these genes represent the level of the labile iron pool (LIP), and their upregulation or downregulation is associated with free iron levels [5,6].

In the literature PCBP1 and PCBP2 proteins are described as iron chaperones that deliver iron to ferritin [7,8]. Thus, an increase in expression of these genes may appear concurrently with a rise in the labile iron pool. However, little is known about how the factors that increase oxidative stress or antioxidant capacity affect mRNA levels of genes associated with iron metabolism.

An augmentation in oxidation accompanies exercise training but also ageing and some diseases. An easy way to increase antioxidant capacity is to supplement antioxidant vitamins such as vitamin C. This vitamin plays a crucial role in preventative therapy against ageing [9] and diseases [10,11]. However the data on its beneficial effects are conflicting and the findings vary with dose, age, and type of exercise [12,13,14].

It is well documented that supplementation with vitamins C and E during acute exercise did not attenuate oxidative stress in muscle cells [15]. Additionally, vitamin C also influences iron uptake by increasing its absorption from the gastrointestinal tract. These data support the close association between vitamin C, iron or ferritin concentration, and exercise.

According to Mankowski et al. [16], antioxidant supplementation can improve training effects by the reduction in exercise production of reactive oxygen species (ROS) but may also impair training adaptation. Moreover, antioxidative supplements could block an increase in exercise level through reduced expression of some genes (such as PGC-1α) associated with ROS-stimulated mitochondrial biogenesis. It is also unclear whether antioxidant supplementation influences a decrease of enzymatic protection against oxidative stress.

Unfortunately, to this day, there are no published data showing the effects of vitamin C supplementation combined with exercise in terms of adaptation of ferritin mRNA in leukocytes. Furthermore, there are no data which could answer the question of whether introducing supplementation of ascorbic acid during the training period is important in regard to iron metabolism. Moreover, in current data, there are many studies associated with antioxidant supplementation, but usually in healthy and active men or as an adjunct to treatment of cancer and other diseases. Thus, we decided to investigate the influence of 1000 mg vitamin supplementation used in elderly (postmenopausal) healthy women.

The aim of this study was to evaluate the effects of sessions of six-week endurance training with and without six-week oral vitamin C supplementation (1000 mg daily) on plasma oxidative stress and antioxidant capacity, as well as on genes associated with cellular stress response and iron metabolism. We assumed that an increase in antioxidant capacity, induced by training and vitamin C supplementation, would have resulted in a decrease in mRNA concentration of the tested genes. Moreover, we assumed that there were no differences in mRNA levels of genes associated with iron metabolism regardless of the timing of supplementation during training.

## 2. Methods

### 2.1. Subjects

The study protocol was completed by 15 women (from the total of 24 who started the trial) with a mean age of 69.2 ± 6.5 years and body weight of 72.9 ± 12.7 kg. We excluded from the analysis the participants who had less than 90% attendance in training. All participants were volunteer students at the University of the Third Age at the Gdansk University of Physical Education and had never trained professionally before. The volunteers had to meet the following inclusion criteria: age between 60 and 80 years, a medical certificate confirming that there were no contraindications to participation in the research, no participation in other projects, a low level of physical activity, and a properly-balanced diet in terms of energy content and basic nutrients. We excluded from the study women who were under 60 or over 80 years of age; had movement limitations that disrupted training or who had ongoing injuries; had endoprostheses or other conditions that limited the possibility of performing tests (including InBody); had poor health in general, including neoplastic diseases, advanced cardio-respiratory diseases, arrhythmias, history of arterial congestion, hypertension (>160/100 mm Hg), transient ischemic attacks, thyroid malfunction, diabetes, were smokers, had high total cholesterol levels (>300 mg/dL), were on a weight-loss diet, or who were taking anti-inflammatory drugs. None of the women had a disease requiring constant treatment and all avoided supplementation for 3 months before the experiment. Additionally, only those women whose dietary intake of energy, protein, fat, and carbohydrates was within the norms established for a given age group for the Polish population qualified to participate in the project.

The women were divided randomly into two groups (simple randomization). Group 1 (n = 7, mean ± SD age of 68.75 ± 7.5 years and body weight of 74.26 ± 16.87 kg) received oral supplementation with vitamin C (1000 mg daily) for the first six weeks of training while group 2 (n = 8, mean ± SD age of 67.7 ± 5.6 years and bodyweight of 71.46 ± 5.39 kg) received cellulose. During the next 6 weeks, the groups received the alternative supplement. The study was double-blinded (products in caps, looking similar). Body mass analysis was performed using the InBody 720 composition analysis (Inbody, Biospace Co. LTD., Seoul, Korea). The anthropometric characteristics of the two tested groups are summarized in Table 1.

### 2.2. Ethics Approval and Consent to Participate

This study was approved by the Bioethics Committee for Clinical Research at the Regional Medical Chamber in Gdansk (permission number KB-10/16, Gdansk, 12.04.2017) and conducted according to the Declaration of Helsinki. All participants gave their written, informed consent prior to participation and were informed about the possibility of withdrawal at any time for any reason.

### 2.3. Diet Analysis

All participants recorded their dietary intake for 5 days—3 working days and 2 holiday days at the beginning of the experiment—before starting the training and vitamin C supplementation. The obtained data were analyzed using the Diet 5.0 software, developed by the Food and Nutrition Institute (Warszawa, Poland). Meal sizes were estimated using the photo album of products and dishes [17]. On this basis, the average amount of energy consumed per day, as well as the percentage of proteins, fats, and carbohydrates, were calculated. Participants were asked not to change their nutritional habits during the study. The study took place between November and February i.e., in the winter season, and there was no seasonal change in nutrition.

### 2.4. Training

The training program was implemented using the basic principles of health-related training. The 15 women participated in a 12-week multidisciplinary training program (2 × 6-week periods, three times per week for 60 min) consisting of water gymnastics, gyrokinesis, stabilization training, and Nordic walking at moderate intensity. Each 6-week training period involved 1080 min of exercise (total duration of 2160 min). During each session, participants maintained their heart rate (Polar H1) at less than 130 beats/min.

### 2.5. Supplementation

In the first 6 weeks of training, group 1 received 1000 mg of Vitamin C (Max VitaC 1000, Colfarm, Poland) and group 2 received cellulose in tablets (Colfarm, Poland). In the following 6 weeks of the training period, participants received the alternative supplement (vitamin C or placebo). Neither they nor the people carrying out the experiment knew what supplements had been given. The choice of vitamin C dose was due to its easy availability and high frequency of consumption in the Polish population.

### 2.6. Report of Determination of VO_2_ Max

The VO_2_ max. was determined directly with the cycloergometer, Ergoline Ergoselect 150p (Jaeger OxyconPro) and a gas analyzer (Jaeger OxyconPro).

Each participant was tested according to the following procedure: 2 min for registration of resting values followed by 5 min of a warm-up with 30 W load and 60 rpm cadence; then the commencement of the appropriate exercise test with a gradual increase of the load by 10 W in intervals of 1 min. Each participant’s test was interrupted when they were exhausted and unable to continue with a given power and the required term of 60 rpm, or after the occurrence of symptoms indicative of the need to end the trial. The results of the VO_2_ max. are presented in Table 1.

### 2.7. Blood Collection, Analysis of Vitamin C Concentration, Total Oxidative Status (TOS)/Total Oxidative Capacity (TOC), Total Antioxidative Status (TAS)/Total Antioxidative Capacity (TAC) Analysis and Genetic Research

Three blood samples (5 mL) were collected from each participant—at baseline and 24 h after the first and second training periods.

### 2.8. Genetic Evaluation

RNA was isolated using the procedure of Chomczynski and Sacchi [18] as described previously by Grzybkowska et al. [6]. Venous blood (2 mL) was collected into special Vancouver tubes with ethylenediaminetetraacetic acid (EDTA) as the anticoagulant. To eliminate erythrocytes, blood samples were treated with red blood cell lysis buffer (RBCL) (A&A Biotechnology, Gdynia, Poland) on ice (20 min) and centrifuged (3000× *g* at 4 °C for 10 min). The obtained leukocytes were lysed using Fenozol (A&A Biotechnology, Gdynia, Poland). After 5 min incubation 200 µL of chloroform (POCH, Gliwice, Poland) was added and the suspension was shaken. The water phase was moved to another Eppendorf tube, and to precipitate RNA, 500 µL of isopropanol (POCH, Gliwice, Poland) was added. Then samples were centrifuged again (10,000× *g*, 30 min, 4 °C). The obtained pellet of RNA was washed two times in 1 mL of 75% ethanol and centrifuged one more time (7500× *g* at 4 °C). Then, ethanol was removed and the pellet was set aside to dry. Dry RNA was diluted in 20 μL molecular grade water. Purity and quality were evaluated spectrophotometrically (BioPhotometer Plus, Eppendorf, Germany). For reverse transcription, 1000 ng of pure RNA (A260/280 ≥ 1.7) was used (AffinityScript QPCR cDNA Synthesis Kit: Agilent, Poland) on Eppendorf Mastercycler Gradient 5331. The profile of the reaction was in accordance with the manufacturer′s instructions. Obtained cDNA was diluted 10-fold indirectly before the PCR reaction. Each gene’s expression was detected using quantitative real-time PCR (qRT-PCR, Aria, Agilent, Department Poland). To amplify gene expression the following primers were used:For *TUBB* (tubulin beta class I, NM_001293213): forward primer: TCCACGGCCTTGCTCTTGTTT and reverse primer: GACATCAAGGCGCATGTGAAC;For *FTH1* (NM_002032): forward primer: TCCTACGTTTACCTGTCCATG and reverse primer: CTGCAGCTTCATCAGTTTCTC;For *FTL* (NM_000146): forward primer: GTCAATTTGTACCTGCAGGCC and reverse primer: CTCGGCCAATTCGCGGAA;For PCBP1 (NM_006196): forward primer: AGAGTCATGACCATTCCGTAC and reverse primer: TCCTTGAATCGAGTAGGCATC;For PCBP2 (NM_001128913): forward primer: TCCAGCTCTCCGGTCATCTTT and reverse primer: ACTGAATCCGGTGTTGCCATG;For *CAT* (NM_001752): forward primer: GATGGACATCGCCACATGAAT and reverse primer: AAGATCCCGGATGCCATAGTC;For *FOXO3A*: forward primer: TTCAAGGATAAGGGCGACAGC and reverse primer: CCCATCAGGGTTGATGATCCA.

### 2.9. Vitamin C Plasma Concentration Measurement

Plasma vitamin C concentrations were determined using the method of Robitaille and Hoffer [19]. In brief, cold trichloroacetic acid (TCA) 20% (0.4 mL) and cold dithiothreitol (DTT) 0.2% (0.4 mL) were added to 0.2 mL of plasma. The samples were then vortexed for 2 min, centrifuged (10,000× *g* at 4 °C for 10 min), and the resultant supernatant was frozen at −80 °C for further analysis. Analysis of vitamin C concentrations was performed in an accredited laboratory in Krakow. Concentrations were measured using high-performance liquid chromatography (HPLC) with UV detection (wavelength: 245 nm). Chromatography was performed on a reverse-phase chromatographic column (RP-18) (Merck, Darmstadt) with a length of 25 cm, diameter 4.6 mm, and a grain diameter of 5 μm, and the mobile phase pH was 2.7. A 2140 Rap1d Spectral Detector Optical Unit UV detector (LKB Bromma, Sweden) and Rheodyne^®^ (Model 7010) injector (Rheodyne, Germany) were used. The retention time of ascorbic acid was 3 min. [19].

### 2.10. TOS/TOC and TAS/TAC Analysis

For TOS/TOC and TAS/TAC analysis, venous blood was collected into BD Vacutainer tubes (Becton Dickinson, Franklin Lakes, NJ, USA) and centrifuged (3000× *g* at 4 °C for 10 min), and the obtained plasma was stored immediately at −80 °C for further analysis. To evaluate TOS/TOC a photometric PerOx assay kit was applied (Immundiagnostik AG, Germany), and for TAS/TAC an ImAnOx assay kit (Immundiagnostik AG, Germany) was applied. Prooxidative/antioxidative balance was also calculated. The photometric PerOx test was applied (Immunodiagnostic, Department Poland); for TAS, ImAnOx (Immunodiagnostic, Department Poland) was applied.

### 2.11. Statistical Analysis

For all results, the presence of a normal distribution was checked with the Shapiro–Wilk test. Results before and after each six-week training period were compared with parametric or non-parametric (Wilcoxon) tests, as appropriate. To evaluate within-person changes, the paired t-test and ANOVA one-way test were applied. The prooxidative/antioxidative balance was calculated as the prooxidative/antioxidative ratio. The genetic data were collected and relative gene expressions were analyzed in Microsoft Excel 2015. The level of mRNA was calculated using the comparative method of Schmittgen and Livak [20]. The mRNA levels of the tested genes were described as the differences in the cycle threshold values normalized to the TUBB mRNA levels, i.e., ΔC_T_ = C_T_ of gene − C_T_ of TUBB. The data were transformed into linear values, and statistical significance was evaluated using the Shapiro–Wilk test to assess for normal distribution and the Wilcoxon test for comparison of results before and after each training period. To determine the significance of differences between the groups, *t*-test and ANOVA two-way were applied. All calculations and graphics were performed using GraphPad Prism 6.0. Differences were considered statistically significant at a level of *p* ≤ 0.05.

## 3. Results

No changes in VO_2_ max. were observed during the experiment, both during the training period supported by supplementation and without supplementation (see Table 2 for reference). Body mass significantly decreased in group 2 after the training period with 1000 mg Vitamin C supplementation, however, in both groups, the tendency to decrease was noted in the second six weeks of training. Fat mass significantly decreased in both groups after 12 weeks of training (compared to baseline). Interestingly, muscle mass significantly decreased in group 2 after 12 weeks of training. No significant differences between groups were observed in tested parameters, independently on supplementation time (first or second six weeks of training, Table 2).

### 3.1. Gene Expression

No significant differences in mRNA levels were observed between groups at baseline. However, mRNA levels of the two ferritin genes significantly decreased after six weeks of training supported by Vitamin C supplementation (in group 1 from 2^64.24 to 11.06, *p* = 0.03 and in group 2 from 2^53.77 to 2^16.03, *p* = 0.03, Figure 1A). At the same time, the decrease in *FTL* mRNA was observed in both tested groups however, significant changes were observed only in group 2 (for group 1 from 2^14.94 to 2^7.64 and for group 2 from 2^18.53 to 2^4.53, *p* = 0.01, Figure 1B). No significant changes were observed for other tested genes, although a slight increase in *PCBP1* and *PCBP2* mRNA and decrease in *FOXO3A* and *CAT* were observed in both groups (Figure 1C–F). There were low levels of *CAT* and *FOXO3a* mRNA throughout the study (Figure 1E,F).

No significant differences in the mRNA of tested genes were observed during six weeks of the unsupplemented training period (Figure 2). At the end of the experiment independently of the time of supplementation (at first or second six weeks of training), there were no differences between groups in RNA levels. As for the effect of supplementation in different periods, the differences in *FTH* mRNA between groups before the unsupplemented period were significant (*p* = 0.01).

### 3.2. Control of Diet and Vitamin C Concentration

Dietary analysis indicated energy shortages in group 2 deviating from standards for the Polish population amounting to 1900 kcal/day. Protein intake accounted for 17% of the caloric value of the diet in both groups and was within the norm of 15–20% of energy. Fat intake differed in both groups from the norm for women with the same body weight and age of 79 g/day. Consumption of carbohydrates in group 1 accounted for 50% of the calorie diet value in group 2 and constituted 47% of the calorific value and was in the norm of 45–65% of energy [21] (Table 3).

Further, the participants achieved the recommended daily intake of vitamin C of around 100 mg per day via consumption of foods like peppers, apples, and oranges. Participants in group 2 had lower consumption of kilocalories per day resulting from a significantly-lower amount of fat in their diet. Both groups had a similar intake of vitamin C per kg of body weight.

The volunteers in group 1 had a mean basal vitamin C plasma concentration of 12.2 mg/L, while those in group 2 had a mean concentration of 14.8 mg/L. All participants had basal vitamin C concentrations within the physiologically normal range (6–20 mg/L). After six weeks of training with supplementation, the vitamin C plasma concentration in both groups increased significantly (in group 1 mean value changed from 12.2 to 14.9 mg/L and in group 2 from 14.8 to 19.5 mg/L, *p* = 0.03). Six weeks of training without vitamin C supplementation did not affect the plasma concentration of vitamin C. There were no significant differences between groups in the plasma vitamin C concentrations during the whole experiment (Figure 3A,B).

### 3.3. TOS/TOC and TAS/TAC Analysis

High levels of total oxidative status were maintained throughout the study in both groups (≥350 umol/L for plasma). Little change in TOS was noted during the experiment, although the mean values after supplementation were slightly higher in both groups (for Group 2 the increase was from 465 to 509 umol/L, Figure 4A). In the supplementation period, TAS/TAC status, as well as prooxidative/antioxidative ratio, remained unchanged (Figure 4B,C). The TAS level in participants was around the average or less of the physiological norm.

During six weeks of training without supplementation, there were also no significant changes in total oxidative/antioxidative status as well in prooxidative/antioxidative ratio (Figure 5A–C).

## 4. Discussion

In our study, we hypothesized that vitamin C would increase antioxidant capacity and simultaneously decrease oxidative stress and that the expression of genes associated with oxidative stress, such as *FTH*, *FTL,* and *FOXO3a* in leukocytes would decrease. Our findings only partially confirmed these hypotheses. First, after vitamin C supplementation, neither group experienced an increase in TAS/TAC, although both showed a tendency to an increase in TOS/TOC. This means that vitamin C could have a prooxidative effect. During the study period, the prooxidative/antioxidative status remained unchanged at a value of 2.0, indicating high oxidative stress according to reference values. At the same time, plasma vitamin C concentration increased as an effect of supplementation. Second, the decrease in mRNA of *FTH* and *FTL* (both significant) as well as of CAT and FOXO3a, and a slight tendency to increase in *PCBP1* and *PCBP2* mRNA caused by vitamin C did not occur when the participants trained without supplementation. Moreover, the results suggest that the influence of vitamin C on genes associated with iron metabolism is much more important than the applied training.

According to the literature, antioxidants may influence both oxidative stress and iron metabolism [22]. The labile iron pool (LIP) is strictly regulated as it is crucial for cell survival, and to prevent iron toxicity [22,23]. One of the most important proteins involved in the prevention of iron toxicity is ferritin, which is encoded by *FTH1* and *FTL* genes. The main function of ferritin is to store iron intracellularly [24], thus it is a key molecule that limits the oxidative stress [3] and is involved in iron homeostasis [25]. Moreover, ferritin gene expression and post-transcriptional regulation of mRNA activity are sensitive to oxidants [23]. Post-transcriptional activity by oxidants is associated with inactivation of IRP1 through reversible oxidation of critical cysteine residues [2].

Some published studies showed a decrease in plasma iron and ferritin after 12 and 32 weeks of Nordic-walking training [26,27]. The findings of these studies indicate that observed changes in iron metabolism are an important part of adaptation to the training. Unfortunately, in our study, we did not observe significant changes in mRNA levels of tested genes after the applied training. Significant decrease in *FTH* and *FTL* mRNA levels occurs only after the training period supported by vitamin C supplementation.

To our knowledge, there are no published studies investigating the effects of training supported by vitamin C supplementation on *FTH*, *FTL*, *FOXO3a, PCBP1, PCBP2,* or *CAT* mRNA levels in human leukocytes. Our results showed that independent of the length of the training period, a significant decrease in *FTH* mRNA was observed after vitamin C supplementation. A decrease in *FTL* mRNA may indicate or lower oxidative stress within cells in subjects after vitamin C supplementation. Genes encoding iron chaperones (*PCBP1* and *PCBP2)* did not change their expression significantly. This may suggest that intracellular free iron level has not changed. Furthermore, *FOXO3a* and *CAT* have apoptosis-regulation properties [28,29]. Recently it has been shown that FOXO3a controls the expression of ferritin and catalase mRNA [30,31], thus the next goal of this study was to determine the effect of training and supplementation on the expression of these genes. Unfortunately, no significant changes were observed for *FOXO3a* and *CAT* mRNA. Although there were no statistically-significant differences, we observed a similar tendency of changes in *CAT* mRNA levels after vitamin C supplementation compared to *FTH* and *FTL*. In summary, the effect of vitamin C supplementation on leukocytes supports the decrease in intracellular oxidative stress (tendency to decrease *FOXO3a* and *CAT* mRNA, and a significant decrease of *FTH* and *FTL* mRNA) and should be considered positive in regard to human health.

Interestingly, a significant difference between groups was observed in *FTH* mRNA levels before the unsupplemented period. This difference could be associated with the fact, that group 1 had foregone vitamin C supplementation for six weeks before going through the unsupplemented period. Moreover, this effect could still be observed at the end of the experiment (after another six weeks). It is possible that the use of antioxidant supplementation has a long-term effect. However, the duration of the supplementation effect requires further research.

### Vitamin C Concentration and Plasma Pro-/Antioxidant Capacity

Plasma vitamin C concentrations of the participants of our study were in the middle and (after the supplementation) upper range of normal. According to Pearson et al. [32], women have higher vitamin C status compared to men. Moreover, these authors postulated that people with higher vitamin C status exhibit lower weight, body mass index and waist circumference and that their indicators of metabolic health were better. In our study, we have observed a significant decrease in body mass after training in both groups and a substantial decrease in mean body mass after supplementation in group 2. However, after the supplementation, there was a tendency towards increased TOS/TOC in both groups. During our study, participants had high oxidative stress and supplementation did not change the prooxidative/antioxidative balance. The same results were obtained by Bunpo and Anthony [33]. These authors evaluated the changes in oxidative status in healthy young people after 12 weeks of moderate exercise (three times a week) supported by vitamin C in doses of 250 and 500 mg per day. They observed no change in TOS/TOC or TAS/TAC and a slight reduction in antioxidative enzymes (including CAT) in erythrocytes. As a result of this, in the presence of Fe^3+^ or Cu^2+^, vitamin C could promote a generation of the reactive oxygen species i.e., OH, O^2−^, H_2_O_2_, and ferryl ion [34]. In the current literature, some published reports have described a prooxidative role of vitamin C [34]. Hydrogen peroxide formation increased under the influence of high dose vitamin C and in the presence of an appropriate concentration of iron ions [35,36]. Further, Tsuma-Kaneko et al. [37] suggested that this could exert anti-cancer effects. On the other hand, in patients with chronic lymphocytic leukaemia, vitamin C in a dose of 1000 mg/day had a positive influence on the incidence of infectious complications [38].

Vitamin C is not the only antioxidant that can manifest antioxidant and prooxidant properties. There are some studies in which both effects were observed, not only in humans but also in other species [39,40]. According to Tofolean et al. [41] another antioxidant, epigallocatechin-3-gallate (EGCG), could influence the prooxidative or antioxidative effects, and these effects were strongly associated with used dose [41]. This study was conducted on Jurkat T-cells in human leukaemia.

Gheorghe et al. [42] postulated that the benefits of antioxidant supplementation are dependent on initial antioxidant concentration. The authors suggested that in low dose, antioxidants may improve liver function and in high dose, substance-specific adverse effect was detected [15]. Similar findings associated with supplementation of another vitamin which is associated with antioxidative action (vitamin D) were reported by Zhang et al. [43]. Authors postulated that the relationship between serum vitamin D and the risk of liver cancer could be inverse.

Our study was conducted on healthy elderly women. Ageing is accompanied by an increase in oxidative stress. However, the interpretation of changes in oxidative stress and its influence on ageing is difficult because increased ROS production is associated with healthy ageing as well as ageing caused by inflammation. In the literature, the results related to oxidative stress and ageing obtained by various authors are inconclusive. Some authors postulated an oxidative damage theory of mammalian ageing [44] and indicated that permanent metabolic slowing accompanied by a reduction in oxidative stress could prevent ageing. Further, a healthy diet and regular exercise can be useful to prevent this kind of ageing. According to Bhatti et al. [45], anti-ageing strategies are mainly focusing on reducing mitochondrial dysfunction and oxidative stress. Thus, lifestyle, diet, supplementation, and physical exercise can influence ageing, e.g., through overexpression of sirtuins, which may be inhibited by oral antioxidant supplementation [46]. El Assar et al. [47] postulated that there are two main mechanisms of ageing-related endothelial dysfunction (a determinant factor for cardiovascular disease and health status in the elderly)—oxidative stress and inflammation. They also indicated that chronic activation of NF-κB and downregulation of sirtuins and SOD2 intensify the cellular response to acute ROS generation. Interventions focused on the recovery of endogenous antioxidant capacity and cellular stress response, rather than exogenous antioxidants, could reverse oxidative stress inflammation. This opinion is consistent with our results. In our study, volunteers using supplementation did not manifest a reduction in oxidative stress. In another study investigating the influence of supplementation on adaptation to training, Yfanti et al. [14] concluded that a dose of 1000 mg of vitamin C daily did not alter endurance training adaptation, which is compatible with our observation. The authors observed an increase in VO_2_ max but no difference in metabolic indicators in muscle cells between the supplemented and unsupplemented group. In our study, there was no significant change in VO_2_ max, but changes in body composition were associated with the length of training.

Thus, in our opinion Vitamin C in a dose of 1000 mg/day did not bring the expected effect in a reduction in oxidative stress and its influence on this parameter was neutral. Despite the influence that vitamin C has on oxidative status, changes in ferritins mRNA levels were observed during the supplementation period, however in group 1 these lasted until the end of training. This could indicate a positive effect of vitamin C supplementation in terms of adaptation to training, but further research is needed to confirm this theory.

## 5. Conclusions

Our findings indicated that six weeks of training supported by daily supplementation of 1000 mg of vitamin C did not influence prooxidative/antioxidative balance but did cause a significant decrease in *FTH* and *FTL* mRNA levels in elderly women. We concluded that vitamin C supplementation had a greater effect on gene expression (and could indirectly indicate lower oxidative stress within the cells), compared with its influence on plasma prooxidative/antioxidative balance. The influence on ferritin mRNA levels could indicate a positive effect of supplementation on adaptation to training in elderly women. Additionally, this effect can be observed during the following six weeks of training. Further, the use of supplementation later in the training period caused major changes in body mass, muscle mass, and *FTL* mRNA, but there were no significant differences between the groups. In our opinion, in the absence of plasma vitamin C levels indicative of deficiency, the need for its supplementation is questionable.

### Study Limitations

Our study has some limitations. We investigated the changes caused by training supported by vitamin C supplementation in healthy, elderly women, and thus our findings cannot be extrapolated with confidence to the women of different ages or males. Moreover, only one dose (1000 mg per day) of vitamin C was used. The effects of supplementation may be dose-dependent.

## Figures and Tables

**Figure 1 ijms-21-06469-f001:**
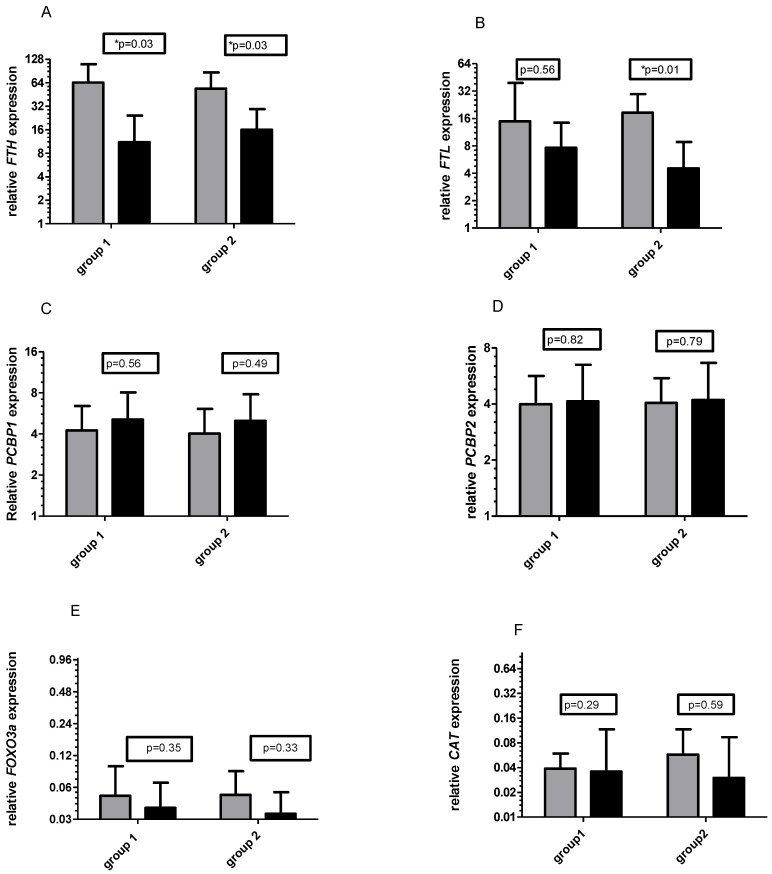
Changes in relative expression (2^) of ferritin heavy chain (*FTH*) (**A**), ferritin light chain (*FTL*) (**B**), poly(rC)-binding protein 1 (*PCBP1*) (**C**), poly(rC)-binding protein 2 (*PCBP2*) (**D**), forkhead box O3A *(FOXO3a*) (**E**), and catalase (*CAT*) (**F**) before (grey bars) and after 6 weeks (dark bars) training period supported by Vitamin C supplementation. All mRNA expressed as 2^ relative expression/TUBB. * significant differences within the group, compared to the baseline value.

**Figure 2 ijms-21-06469-f002:**
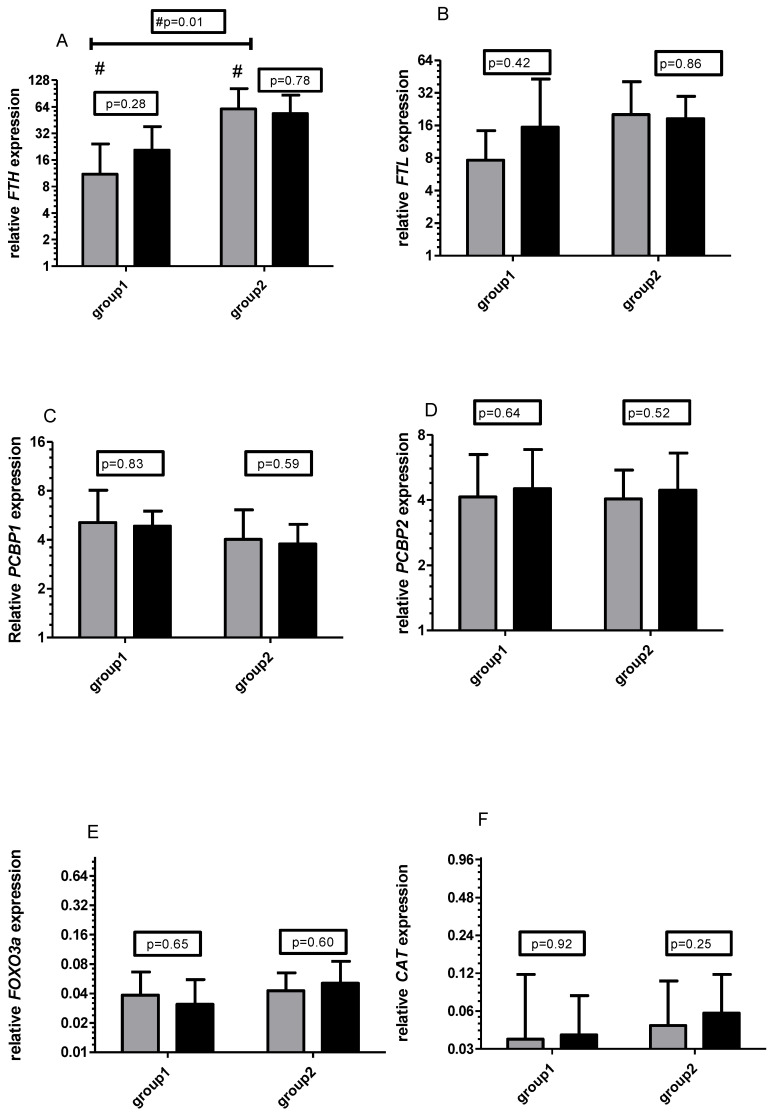
Changes in relative expression (2^) of *FTH* (**A**), *FTL* (**B**), *PCBP1* (**C**), *PCBP2* (**D**), *FOXO3a* (**E**), and *CAT* (**F**) before (grey bars) and after 6 weeks (dark bars) unsupplemented training period. # significant differences between groups. All mRNA expressed as 2^ relative expression/TUBB.

**Figure 3 ijms-21-06469-f003:**
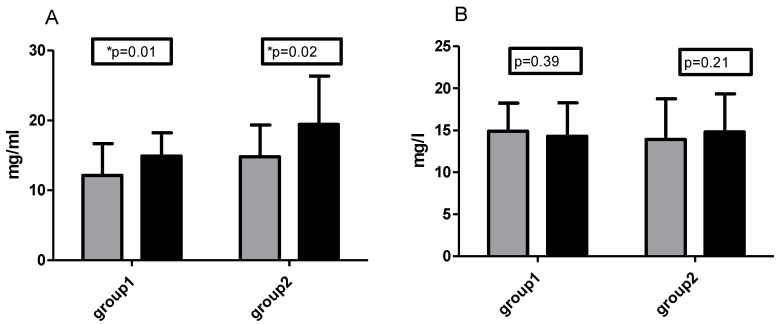
Changes in Vitamin C concentration before (grey bars) and after 6 weeks of training (dark bars). (**A**) Training period supported by supplementation and (**B**) unsupplemented training period. * significant differences compared to the baseline value (*p* ˂ 0.05).

**Figure 4 ijms-21-06469-f004:**
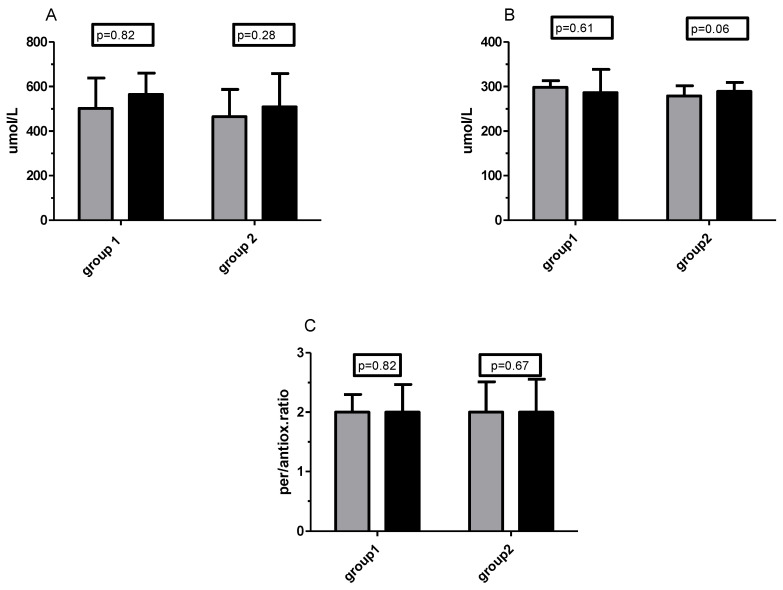
Changes in total oxidative status (TOS)/total oxidative capacity (TOC) (**A**), total antioxidative status (TAS)/total antioxidative capacity (TAC) (**B**), and prooxidative/antioxidative ratio (**C**) before (grey bars) and after 6 weeks of training (dark bars) supported by Vitamin C supplementation.

**Figure 5 ijms-21-06469-f005:**
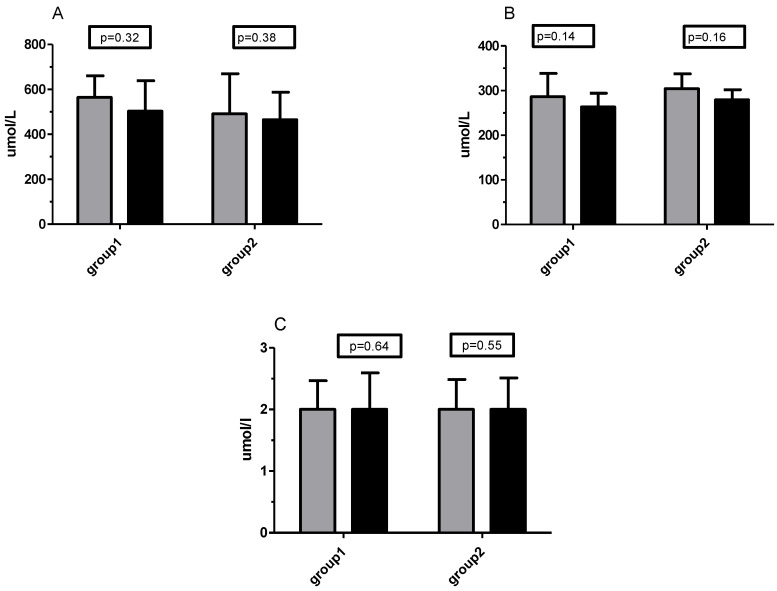
Changes in TOS/TOC (**A**), TAS/TAC (**B**), and prooxidative/antioxidative ratio (**C**) before (grey bars) and after 6 weeks of the unsupplemented training period (dark bars).

**Table 1 ijms-21-06469-t001:** Anthropometric characteristics of participants at baseline (n = 15).

Parameter	I (Baseline)		
	Group 1	Group 2	*p*
**VO_2_ max. (mL/kg/min)**	20.69 ± 4.05	19.73 ± 2.45	0.61
**Body mass (kg)**	74.26 ± 16.87	71.46 ± 5.39	0.68
**Fat (kg)**	29.86 ± 13.86	29.74 ± 3.12	0.98
**Muscle mass (kg)**	24.07 ± 3.76	22.46 ± 2.17	0.34

**Table 2 ijms-21-06469-t002:** Anthropometric characteristics of participants after 6 (II) and 12 (III) weeks of training (*n* = 15).

Parameter		II	*p* (I vs. II)	III	*p* (I vs. III)	*p* (II vs. III)
**VO_2_ max.** (mL/kg/min)	Group 1	20.96 ± 4.47	0.67	21.63 ± 4.66	0.13	0.49
	Group 2	19.62 ± 2.04	0.79	20.62 ± 1.48	0.30	0.27
	***p* (Group 1 vs. Group 2)** = 0.51			***p* (Group 1 vs. Group 2)** = 0.62		
**Body mass** (kg)	Group 1	73.33 ± 16.15	0.55	71.79 ± 16.24	0.13	0.07
	Group 2	70.07 ± 7.12	0.38	65.50 ± 6.95 *^,#^	0.03	0.04
	***p* (Group 1 vs. Group 2)** = 0.63			***p* (Group 1 vs. Group 2)** = 0.36		
**Fat** (kg)	Group 1	26.87± 10.67	0.14	24.98 ± 14.37 *	0.02	0.45
	Group 2	26.60 ± 6.18	0.12	23.07 ± 7.68 *	0.02	0.13
	***p* (Group 1 vs. Group 2)** = 0.95			***p* (Group 1 vs. Group2)** = 0.75		
**Muscle mass** (kg)	Group 1	25.58 ± 4.79	0.55	25.89 ± 4.75 *	0.04	0.85
	Group 2	23.57 ± 2.24	0.07	23.07 ± 1.91	0.45	0.45
	***p* (Group 1 vs. Group 2)** = 0.60			***p* (Group 1 vs. Group 2)** = 0.15		

Baseline values are summarized in Table 2 (I). II—24 h after 6 weeks of training and III—24 h after 12 weeks of training. Values are means (±SD); * significant differences between I and III; # significant differences between II and III.

**Table 3 ijms-21-06469-t003:** Average daily energy values and nutrients of the female diet.

Diet Analysis	Group 1	Group 2
**Energy (kcal)**	2069 ± 702	1634 ± 245
**Proteins (g/day)**	90 ± 32	71 ± 20
**Fat (g/day)**	75 ± 29	62 ± 12
**Carbohydrates (g/day)**	260 ± 130	196 ± 22
**Vit C (mg)**	154 ± 105	141 ± 56
**Vi C (mg/kg b. m.)**	2.13 ± 1.5	2.00 ± 0.79

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
