# Peer review of "Exercise Training and Vitamin C Supplementation Affects Ferritin mRNA in Leukocytes without Affecting Prooxidative/Antioxidative Balance in Elderly Women"

_ijms, 2020, doi:10.3390/ijms21186469_

Round 1
Reviewer 1 Report
This manuscript is on an interesting and relevant topic, but is in need of significant change in the presentation before considered publication. Revisions are necessary before a full evaluation can be made, but a non-exhaustive list of concerns (to start the revision process) is listed below.
- The introduction is not well organized and needs some more defined structure. Break separate topics into distinct paragraphs. Also, some overly repeated words need to be avoided, as general improvements in the language.
- In the Methods, how was randomization performed? Were demographics included in the randomization method?
- Was this clinical trial registered? Did any participants drop from the study? What were the trial exclusions?
- When was the dietary analysis performed in the study? Was it performed at baseline or during the intervention?
- It is strongly suggested that Table 1 is split in two - one with baseline demographics and the other with intervention data.
- More details are needed on the HPLC method for vitamin C analysis - what is the pH of the mobile phase? What wavelength was used for detection? What is the retention time of the ascorbic acid detected?
- Figures in the Results need more descriptive vertical axes, or some other annotation to improve readability. It is difficult to interpret without these additions. The units displayed for some of the genetic tests are cumbersome.
- It is unclear why Group 1 and Group 2 are separated throughout the manuscript without grouping by phase of study/intervention.
- Overall, a better organization of the data is needed with accompanying discussion. It is difficult to tease out the effects from the intervention(s) in its current form.
- Figure 4 is missing some group designations. It is unclear if anything is marked as significant.
Author Response
Response to Reviewer 1.
First, I would like to thank the reviewer for all comments that have contributed to the improvement of our manuscript. We made some changes, according to the Reviewer’s suggestions.
In detail:
1. The Reviewer wrote: „The introduction is not well organized and needs some more defined structure. Break separate topics into distinct paragraphs. Also, some overly repeated words need to be avoided, as general improvements in the language”.
Response: In the present version, the introduction section is clearer, has a more defined structure. A linguistic revision of the manuscript was made.
2. The Reviewer wrote: „In the Methods, how was randomization performed? Were demographics included in the randomization method?”
Response: We have added to method section information, that simple randomization was applied when divided into groups. Group selection was independent of demographic factors.
3. The Reviewer wrote: „Was this clinical trial registered? Did any participants drop from the study? What were the trial exclusions?”
Response: We have been trying to register the study for a year in ANZCTR (Request Id: 377628). Data for last and final approval: 18.03.2020. Unfortunately, until now we are waiting for the final registration (because of the COVID-19 pandemic, studies that are being conducted on COVID-19 have higher priority). Nine of the women had dropped from the experiment and in the present version, we have added exclusion criteria in method section:
„The study protocol was completed by 15 women (from a total of 24 who started the trial but did not complete it)…. We excluded from the analysis the participants who had less than 90% attendance in training. „The volunteers had to meet the following inclusion criteria: age between 60-70 years, a medical certificate confirming that there are no contraindications to participate in the research, no participation in other projects, low level of physical activity and a properly balanced diet in terms of energy content and basic nutrients. We excluded from the study women who: were under 60 or over 70 years of age, had movement limitations that disrupt training, ongoing injuries people with endoprostheses or other conditions that limit the possibility of performing tests (including In Body) Poor health in general, including neoplastic diseases, advanced cardio-respiratory diseases, arrhythmias, history of arterial congestion, hypertension (> 160/100 mm Hg), transient ischemic attacks, thyroid malfunction, diabetes, smoking, total content cholesterol (> 300 mg / dL), weight loss diet, taking anti-inflammatory drugs”. Additionally, only those women whose dietary intake of energy, protein, fat and carbohydrates were within the norms established for a given age group for the Polish population were qualified to participate in the project.
4. The Reviewer wrote: „When was the dietary analysis performed in the study? Was it performed at baseline or during the intervention?”
Response: In the present version in method section, we added this sentence: All participants recorded their dietary intake for 5 days: 3 working days and 2 holiday days at the beginning of the experiment – before starting the training and supplementation of vitamin C.
5. The Reviewer wrote: „It is strongly suggested that Table 1 is split in two - one with baseline demographics and the other with intervention data.”
Response: Now Table 1 includes only parameters at baseline and is presented in the methods section. In the Results section, we presented Table 2 containing parameters after the intervention.
6. The Reviewer wrote: „More details are needed on the HPLC method for vitamin C analysis - what is the pH of the mobile phase? What wavelength was used for detection? What is the retention time of the ascorbic acid detected?”
Response: We added more details on the HPLC method. “Analysis of vitamin C concentrations was performed in an accredited laboratory in Krakow. Concentrations were measured using high-performance liquid chromatography (HPLC) with UV detection (wavelength: 245 nm). Chromatography was performed on a reverse-phase chromatographic column (RP-18) (Merck, Darmstadt) with a length of 25 cm, diameter 4.6 mm and a grain diameter of 5 μm, mobile phase pH was 2.7. 2140 Rap1d Spectral Detector Optical Unit UV detector (LKB Bromma, Sweden) and Rheodyne® (Model 7010) injector (Rheodyne, Germany) were used. The retention time of ascorbic acid was 3 min. [19].
7. The Reviewer wrote: „Figures in the Results need more descriptive vertical axes, or some other annotation to improve readability. It is difficult to interpret without these additions. The units displayed for some of the genetic tests are cumbersome”.
Response: In the current version of the manuscript, all figures are changed and we hope that vertical axes are more descriptive.
8. The Reviewer wrote: „It is unclear why Group 1 and Group 2 are separated throughout the manuscript without grouping by phase of study/intervention”.
Response: We have decided to divide our results according to the supplemented or unsupplemented period. In our opinion, it is now easier to trace, and the comparison between changes caused by training with or without Vitamin C supplementation (supplemented period: I and II in Group 1 and II and III in group 2) is clearly presented.
9. The Reviewer wrote: „Overall, a better organization of the data is needed with accompanying discussion. It is difficult to tease out the effects from the intervention(s) in its current form.
Response: In the current version, we have tried to organize the presented data as suggested.
10. The Reviewer wrote: „Figure 4 is missing some group designations. It is unclear if anything is marked as significant.”
Response: All figures in the present version are corrected.
Thank you very much again for all comments. We hope that our amendments will meet with your approval.
Kind regards, Małgorzata Żychowska & Jędrzej Antosiewicz
Reviewer 2 Report
This is an interesting paper but at this point, it requires a revision:
- The manuscript needs a moderate polishing of the English language.
- Please add the approval number of the ethics committee or council that endorsed the study.
- The authors give no reason why only elderly women were included in the research. This is a selection bias. What was the reason for this selection criterion?
- Please add inclusion and exclusion criteria.
- Please add p-values in the tables and figures. You say you have conducted a statistical analysis of your data but the numbers are missing. Statements must be justified by data.
- Were the subjects healthy? Did they suffer from any health conditions that could have affected their oxidant/antioxidant status? Hypertension? Dyslipidemia? Obesity? Diabetes? Oxidative stress is involved in all of these disorders, as well as a myriad of conditions.
- Healthy ageing is also associated with increased oxidative stress levels. The authors should also acknowledge this in their research and analyze their data according to literature findings on the topic.
- Vitamins are generally potent antioxidants. The authors should review their anti-oxidative stress properties. Please see:
https://www.hindawi.com/journals/omcl/2014/158135/
https://www.sciencedirect.com/science/article/abs/pii/S104366181500290X
https://www.tandfonline.com/doi/abs/10.1080/01635581.2020.1797127
https://doi.org/10.37358/RC.19.2.6977
Author Response
Response to Reviewer 2.
First, I would like to thank the reviewer for all comments that have contributed to the improvement of our manuscript. We made some changes, according to the Reviewer’s suggestions.
In detail:
- The Reviewer wrote: „The manuscript needs a moderate polishing of the English language.”
Response: We have performed the English correction as suggested.
- The Reviewer wrote: „Please add the approval number of the ethics committee or council that endorsed the study."
Rsponse: We have added the number of permission in Ethics section (number of permission KB-10/16
- The Reviewer wrote: „The authors give no reason why only elderly women were included in the research. This is a selection bias. What was the reason for this selection criterion?”
Response: In the present version of the manuscript we have tried to explain selection criteria. We have added in Introduction: “Moreover, in current data, there are many studies associated with antioxidants supplementation, but usually in healthy and active men or as an adjunct to cancer and other diseases treatment. Thus, we decided to investigate the influence of 1000 mg vitamin supplementation used in elderly (postmenopausal) healthy women”.
- The Reviewer wrote: „Please add inclusion and exclusion criteria”.
Response: We have added inclusion and exclusion criteria. This fragment can be found in the Methods section: „The volunteers had to meet the following inclusion criteria: age between 60-80 years, a medical certificate confirming that there are no contraindications to participate in the research, no participation in other projects, low level of physical activity and a properly balanced diet in terms of energy content and basic nutrients. We excluded from the study women who: were under 60 or over 80 years of age, had movement limitations that disrupt training, ongoing injuries people with endoprostheses or other conditions that limit the possibility of performing tests (including In Body).
- The Reviewer wrote: „Please add p-values in the tables and figures. You say you have conducted a statistical analysis of your data but the numbers are missing. Statements must be justified by data.”
Response: In the present version tables and figures were changed according to the Reviewer’s suggestion.
- The Reviewer wrote: „Were the subjects healthy? Did they suffer from any health conditions that could have affected their oxidant/antioxidant status? Hypertension? Dyslipidemia? Obesity? Diabetes? Oxidative stress is involved in all of these disorders, as well as a myriad of conditions.”
Response: All women included in our experiment were healthy and this criterium was very difficult to reach (this is the reason for the low number of participants. In the manuscript we have added a fragment that treats about the exclusion criteria: Poor health in general, including neoplastic diseases, advanced cardio-respiratory diseases, arrhythmias, history of arterial congestion, hypertension (> 160/100 mm Hg), transient ischemic attacks, thyroid malfunction, diabetes, smoking, total content cholesterol (> 300 mg / dL), weight loss diet, taking anti-inflammatory drugs.”
- The Reviewer wrote: „Healthy ageing is also associated with increased oxidative stress levels. The authors should also acknowledge this in their research and analyze their data according to literature findings on the topic”.
Response: In the present version we have discussed oxidative stress and aging more in-depth. We have added: “Our study was conducted on healthy elderly women. Ageing is accompanied by an increase in oxidative stress. However, the interpretation of changes in oxidative stress and its influence on ageing is difficult, because increased ROS production is associated with healthy ageing as well as ageing caused by inflammation. In the literature, the results related to oxidative stress and ageing obtained by various authors are inconclusive. Some authors postulated oxidative damage theory of mammalian ageing [43] and indicated that permanent metabolic slowing accompanied by a reduction in oxidative stress could prevent ageing. Further, a healthy diet and regular exercise can be useful to prevent this kind of ageing. According to Bhatti et al. [44] anti-ageing strategies are mainly focusing on reducing mitochondrial dysfunction and oxidative stress. Thus, lifestyle, diet, supplementation and physical exercise can influence ageing, e.g. through overexpression of sirtuins, which may be inhibited by oral antioxidant supplementation [45]. El Assar et al. [46] postulated that there are two main mechanisms of ageing-related endothelial dysfunction (a determinant factor for cardiovascular disease and health status in the elderly): oxidative stress and inflammation. They also indicated that chronic activation of NF-kB and downregulation of sirtuins and SOD2 intensify the cellular response to acute ROS generation. Interventions focused on the recovery of endogenous antioxidant capacity and cellular stress response, rather than exogenous antioxidants, could reverse oxidative stress inflammation. This opinion is consistent with our results. In our study, volunteers using supplementation did not manifest a reduction in oxidative stress”.
Literature:
- Redman L.M., Smith S.R., Burton J.H., Martin C.K., Il'yasova D., Ravussin E.
Metabolic Slowing and Reduced OxidativeDamage with Sustained Caloric Restriction Support the Rate of Living and Oxidative Damage Theories of Aging. Cell Metab. 2018, 27(4), 805-815. - Bhatti J.S., Bhatti G.K., Reddy P.H.
Mitochondrialdysfunction and oxidative stress in metabolic disorders - A step towards mitochondria based therapeutic strategies. Biochim Biophys Acta Mol Basis Dis. 2017, 1863(5), 1066-1077. - WegmanP., Guo M.H., Bennion D.M., Shankar M.N., Chrzanowski S.M., Goldberg L.A., Xu J., Williams T.A., Lu X., Hsu S.I., Anton S.D., Leeuwenburgh C., Brantly M.L. Practicality of intermittent fasting in humans and its effect on oxidative stress and genes related to aging and metabolism. Rejuvenation Res. 2015, 18(2), 162-172.
- El Assar M., Angulo J., Rodríguez-Mañas L. Oxidative stress and vascular inflammation in aging Free Radic Biol Med. 2013, 65, 380-401.
8. The Reviewer wrote: ”Vitamins are generally potent antioxidants. The authors should revied their anti-oxidative stress properties. Please see: https://www.hindawi.com/journals/omcl/2014/158135/
https://www.sciencedirect.com/science/article/abs/pii/S104366181500290X
https://www.tandfonline.com/doi/abs/10.1080/01635581.2020.1797127
https://doi.org/10.37358/RC.19.2.6977”
Response: There are very interesting papers. We have included these articles in the manuscript in the Introduction and the Discussion section:
Introduction: „For example, Gheorghe et al. [15] postulated that the benefits of antioxidant supplementation are dependent on antioxidant concentration. The authors suggested that in low dose, antioxidants may improve liver function and in high dose, adverse effect was detected [15]. Similar findings associated with supplementation by another vitamin which is associated with antioxidative action (vitamin D) was reported by Zhang et al. [16]. Authors postulated that the relationship between serum vitamin D and the risk of liver cancer could be inverse.”
Discussion: On the other hand, in some types of cancer, vitamin C in dose 1000 mg/day had a positive influence on the incidence of infectious complications in patients with chronic lymphocytic leukaemia [41].
Vitamin C is not the only antioxidant, which can manifest antioxidant and prooxidant properties. According to Tofolean et al. [42] another antioxidant, (epigallocatechin-3-gallate (EGCG)), in human leukaemia, Jurkat T-cells could influence on the oxidative or antioxidative effects and these effects were strongly associated with used dose [42]. This study was conducted Jurkat T-cells in human leukemia.
Thank you very much again for all comments. We hope that our amendments will meet with your approval.
Kind regards,
Małgorzata Żychowska & Jędrzej Antosiewicz
Round 2
Reviewer 1 Report
In its revised form, the manuscript is improved. There are a few changes to the text that are necessary before publication, and a few suggestions for improvements:
Necessary changes:
- Please check the vitamin C concentrations throughout. 12 ug/dL is equal to approximately 0.68 umol/L - much below the level of deficiency. Typical reference ranges for plasma vitamin C in adults are between 0.4 or 0.6-1.5 mg/dL
- Please change the Y axes in Figures 1 and 2 to their gene names, such as "Relative FTH Expression" and elaborate in the figure legend, such as "All mRNA expressed as 2^relative expression/TUBB." This improves readability greatly.
- Reformat tables to make sure all numbers fit on a single line (and are not line-wrapped). Be consistent with the significant digits within the level of precision; two decimal places are not necessary for values above 1.0
- Spaces appear throughout the manuscript text, seemingly at random - these need to be removed.
- Make figure sizes consistent (see Figure 4); Make figure axes consistent wherever possible (see Figure 3)
Suggestions:
- The changed portion introduction speaks about liver function and antioxidant supplementation - this seems out of place for this section and may be better moved to the discussion. It is suggested that the authors instead describe the large body of work around vitamin C and/or vitamin E supplementation and exercise (and the controversies surrounding antioxidant supplementation and exercise adaptation)
- Please evaluate the data in regard to these studies mentioned in the previous suggestion - does any of your data support vitamin C supplementation reducing adaptations or exercise, or is it neutral?
- The FTH mRNA levels are different between group 1 and group 2 at the start of their unsupplemented period (Figure 2, panel A). Group 1 did not see any significant rise in FTH mRNA during the unsupplemented period. Since group 1 unsupplemented period was after their supplemented period this would suggest that the changes in FTH mRNA persist after supplementation ends (at least for this 6 week interval?). If this is the case, the authors should speculate on this lasting effect in the Discussion.
- If the data for the supplementation periods were combined, is there any significant effect of vitamin C supplementation on any of the markers? In particular, the effects on CAT mRNA look to be changing, but it is hard to judge the variability. Similarly, would TOS/TOC ratio also approach significance?
Author Response
Response to Reviewer 1.
First, I would like to thank the reviewer for all comments that have contributed to the improvement of our manuscript. We made some changes, according to the Reviewer’s suggestions.
Necessary changes:
1. The Reviewer wrote: “Please check the vitamin C concentrations throughout. 12 ug/dL is equal to approximately 0.68 umol/L - much below the level of deficiency. Typical reference ranges for plasma vitamin C in adults are between 0.4 or 0.6-1.5 mg/dL”.
Response: Thank you very much for your exact revision. Thank you for pointing out this mistake in the vitamin C concentration unit in our manuscript. They were given in [ug/dl], while the correct unit should be [mg/l]. This error has been corrected in the text.
2. The Reviewer wrote: “Please change the Y axes in Figures 1 and 2 to their gene names, such as "Relative FTH Expression" and elaborate in the figure legend, such as "All mRNA expressed as 2^relative expression/TUBB." This improves readability greatly”.
Response: In present version of our manuscript we changed Figures according to suggestion.
3. The Reviewer wrote: “Reformat tables to make sure all numbers fit on a single line (and are not line-wrapped). Be consistent with the significant digits within the level of precision; two decimal places are not necessary for values above 1.0”.
Response: In present version we corrected all numbers fit.
4. The Reviewer wrote: ”Spaces appear throughout the manuscript text, seemingly at random - these need to be removed”.
Response: We checked once again and removed spaces.
5. The Reviewer wrote: „Make figure sizes consistent (see Figure 4); Make figure axes consistent wherever possible (see Figure 3)”.
Response: we provided suggesting changes in present version of our manuscript.
Suggestions:
Thank you very much for all suggestion. In present version we did following changes:
1. The Reviewer wrote: “The changed portion introduction speaks about liver function and antioxidant supplementation - this seems out of place for this section and may be better moved to the discussion. It is suggested that the authors instead describe the large body of work around vitamin C and/or vitamin E supplementation and exercise (and the controversies surrounding antioxidant supplementation and exercise adaptation)”.
Response: Suggested fragment we moved to the discussion section. In the Introduction we added following fragment: „According to Mankowski et al. [16] antioxidant supplementation can improve training effects by reduction exercise production of ROS but also may impair training adaptation. Moreover, antioxidative supplements could block an increase in exercises possibilities throught reduced in expression some genes (such as PGC-1α) associated with ROS-stimulated mitochondrial biogenesis. It is also unclear, if antioxidant supplementation influence on decrease in production enzymatic protection against oxidative stress.”
Additionaly we added: It is also possible that high dose of antioxidant vitamin in elderly people may especially improve metabolic status, including poor physical condition. justifying the group selection.
2. The Reviewer wrote: „Please evaluate the data in regard to these studies mentioned in the previous suggestion - does any of your data support vitamin C supplementation reducing adaptations or exercise, or is it neutral?”
Response: In present version we added the fragment in discussion and conclusion:
In discussion: Thus, in our opinion Vitamin C in dose 1000mg/day did not bring the expected effect in reduction in oxidative stress and its influence on this parameters were neutral. Despite the ifluence Vitamin C on oxidative status changes in ferritins mRNA observed only during supplementation period could indicated positive effect of this supplementation in terms of adaptation to training, but further research is needed to confirmations or rebuttals this theory.
In conclusion: The influence on ferritins mRNA can indicate positive effect of supplementation on adaptation to training in elderly women.
3. The Reviewer wrote: „The FTH mRNA levels are different between group 1 and group 2 at the start of their unsupplemented period (Figure 2, panel A). Group 1 did not see any significant rise in FTH mRNA during the unsupplemented period. Since group 1 unsupplemented period was after their supplemented period this would suggest that the changes in FTH mRNA persist after supplementation ends (at least for this 6 week interval?). If this is the case, the authors should speculate on this lasting effect in the Discussion.”
Response: Thank you very much for this comments. In present version we added:
In discussion: Interesting, significant difference between groups was observed in FTH mRNA before un-supplemented period. This significant difference could be associated with fact, that Group 1 was in this time after supplementation (supplementation in the first 6 weeks of training). Moreover, at the end of experiment this effect was still lingered. It is possible that the use of antioxidant supplementation allows for a long-term effect (in our experiment another 6 weeks) .However, the duration of the supplementation effect requires further research.
In conclusion: Additionally, this effect can be observed during next 6 weeks of training.
4. The Reviewer wrote: „If the data for the supplementation periods were combined, is there any significant effect of vitamin C supplementation on any of the markers? In particular, the effects on CAT mRNA look to be changing, but it is hard to judge the variability. Similarly, would TOS/TOC ratio also approach significance?”
Response: We combine data according to supplemented and unsupplemented period during developing statistics. The significants was only for Ferritin H mRNA: p=0.001. For anothers parameters there were no significant changes: for FTL p=0.07, for CAT p=0.4, for TOS/TOC p=0.36. Unfortunatelly, SD values were affected the lack of significance.
Thank you very much again for all comments. We hope that our amendments will receive approval.
Kind regards, Małgorzata Żychowska & Jędrzej Antosiewicz
Reviewer 2 Report
The authors have accurately addressed my suggestions and now the paper is suitable for publication.
Author Response
Response to Reviewer.
Thank you very much for your valuable comments and help in improvement our article. Kind regards, Małgorzata Żychowska & Jędrzej Antosiewicz